# Recent Advances in Micro-LEDs Having Yellow–Green to Red Emission Wavelengths for Visible Light Communications

**DOI:** 10.3390/mi14020478

**Published:** 2023-02-18

**Authors:** Konthoujam James Singh, Wei-Ta Huang, Fu-He Hsiao, Wen-Chien Miao, Tzu-Yi Lee, Yi-Hua Pai, Hao-Chung Kuo

**Affiliations:** 1Department of Photonics, College of Electrical and Computer Engineering, National Yang Ming Chiao Tung University, Hsinchu 30010, Taiwan; 2Semiconductor Research Center, Hon Hai Research Institute, Taipei 11492, Taiwan; 3Department of Electrophysics, College of Science, National Yang Ming Chiao Tung University, Hsinchu 30010, Taiwan

**Keywords:** micro-LEDs, visible light communication, modulation bandwidth, high speed

## Abstract

Visible light communication (VLC), which will primarily support high-speed internet connectivity in the contemporary world, has progressively come to be recognized as a significant alternative and reinforcement in the wireless communication area. VLC has become more popular recently because of its many advantages over conventional radio frequencies, including a higher transmission rate, high bandwidth, low power consumption, fewer health risks, and reduced interference. Due to its high-bandwidth characteristics and potential to be used for both illumination and communications, micro-light-emitting diodes (micro-LEDs) have drawn a lot of attention for their use in VLC applications. In this review, a detailed overview of micro-LEDs that have long emission wavelengths for VLC is presented, along with their related challenges and future prospects. The VLC performance of micro-LEDs is influenced by a number of factors, including the quantum-confined Stark effect (QCSE), size-dependent effect, and droop effect, which are discussed in the following sections. When these elements are combined, it has a major impact on the performance of micro-LEDs in terms of their modulation bandwidth, wavelength shift, full-width at half maximum (FWHM), light output power, and efficiency. The possible challenges faced in the use of micro-LEDs were analyzed through a simulation conducted using Crosslight Apsys software and the results were compared with the previous reported results. We also provide a brief overview of the phenomena, underlying theories, and potential possible solutions to these issues. Furthermore, we provide a brief discussion regarding micro-LEDs that have emission wavelengths ranging from yellow–green to red colors. We highlight the notable bandwidth enhancement for this paradigm and anticipate some exciting new research directions. Overall, this review paper provides a brief overview of the performance of VLC-based systems based on micro-LEDs and some of their possible applications.

## 1. Introduction

Micro-light-emitting diodes (micro-LEDs) have drawn a great deal of interest as a potential candidate for the next-generation visible light communication (VLC) research community due to their high modulation bandwidth, low power consumption, array integration ability, ability to multiplex wavelengths, and compatibility with current solid-state lighting systems [1,2,3]. In order to enable the production of networked, portable, high-speed communication analogous to Wi-Fi, VLC—also known as Li-Fi—is a wireless technology that utilizes light emitted by LEDs. It can address a number of issues with the current RF technology, making it a possible replacement for radio frequency (RF) or cellular network communication [2,4,5]. VLC can fill in the gap for full spectrum communication following the introduction of high-performance LEDs technology [6]. Using visible LEDs as transmitters does not necessitate any specific power limits, unlike the conventional free space optical communication technology (which often uses the infrared spectrum). It can assure data security, make system integration easier, and help with the increasingly significant problem of the limitation of the spectra [6]. The use of VLC based on micro-LEDs has advanced tremendously in recent years. These devices have displayed outstanding performance owing to their benefits of good heat dissipation, uniform current distribution, and small RC constant [7,8,9]. However, its application in long distance communication is limited by the low light output power caused by the small luminous effective area of the device. VLC has the unique feature of being able to realize displays and illumination at the same time. White light-emitting diodes (WLEDs) are regarded as an effective, energy-saving solid-state light source and have attracted a lot of interest because of their high efficiency and environmental protection [10,11,12]. The LEDs’ on/off switching period should be as short as possible to accurately transmit high speed data. Consequently, the device must have a short fluorescence lifetime. Due to their smaller size, higher injection current density, and lower RC time constant than conventional LEDs, micro-LEDs offer a higher modulation bandwidth, which greatly increases the transfer rate of VLC systems [13,14,15]. Due to their small size and ease of integration, micro-LED-based VLC systems give wearable devices additional special advantages [16,17]. Furthermore, due to their light sensitivity, micro-LEDs may now be used to create high-performance receivers, opening up new opportunities for their implementation.

The frequency response of LEDs is influenced by the carrier dynamics and parasitic geometrical capacitance of the LEDs, characterized by the carrier lifetime, *τ_n_*, and resistance–capacitance (RC) time constant, *τ_RC_*, respectively. The parasitic capacitance can be neglected if a small forward bias is applied and the active region of the device is small. The modulation frequency at which the LED power transfer efficiency is lowered by 3 dB is known as the LED 3 dB modulation optical bandwidth, which is expressed in Refs. [18,19,20,21].
(1)$$\;f_{3dB}=\frac{\sqrt{3}}{2\pi \tau }$$

The modulation bandwidth of LEDs is governed by the total response time, *τ*, which consists of two parts: the RC time constant, *τ_RC_*, and the carrier lifetime, *τ_n_*. As a result, the modulation bandwidth can be significantly enhanced by lowering the total response time and, hence, a higher channel capacity can be obtained for the overall system. The carrier lifetime can be expressed as
(2)$$\frac{1}{\tau_{n}}=\frac{1}{\tau_{r}}+\frac{1}{\tau_{nr}}$$
where *τ_r_* and *τ_nr_* denote the radiative and nonradiative recombination lifetime, respectively. Hence, it is possible to optimize the area of the device-active region, which will undoubtedly affect the carrier lifetime, and thus control the modulation bandwidth of LEDs. The existence of the quantum-confined Stark effect (QCSE) and large RC time constant in traditional LEDs limits their potential for VLC applications [22,23,24]. Consequently, micro-LEDs are preferred to the conventional LEDs as the transmitter for the optical communication system. Additionally, due to their small size, micro-LEDs can attain higher current densities and a shorter RC time constant, leading to a higher modulation bandwidth. Moreover, the enormous potential of micro-LEDs in the display industry gives us hope that this transmitter can be employed for applications combining display communication and wireless transmission. At present, high-performance GaN/InGaN blue micro-LEDs with a variety of structural configurations have been extensively studied in terms of efficiency or bandwidth [25,26,27]. The design of an ultra-thin quantum well (QW) structure with an InGaN layer of 1 nm and the use of optimal variable temperature growth procedures enabled the achievement of a high modulation bandwidth of 1.53 GHz [28]. Lin et al. reported a 10 µm c-plane green micro-LED achieving a record modulation bandwidth of 1.31 GHz at a current density of 41.4 kA/cm^2^ and can achieve a maximum data rate of 5.789 Gbps in series combination after combined with an orthogonal frequency division multiplexing modulation scheme [29]. In another context, Shan et al. made the first demonstration of a 20 m blue micro-LED that is constructed on a 2-inch freestanding c-plane GaN substrate and has a high bandwidth of over 1 GHz [30]. GaN-on-Si green micro-LEDs with high bandwidth of up to 613 MHz and a data rate of 4.65 Gbps were recently reported by Zhu et al. in a nondestructive transfer printing technique and performance characterization [31]. Furthermore, in terms of free-space VLC and underwater optical wireless communication, the performance of such GaN blue micro-LEDs is significantly superior to that of long-wavelength LEDs [32,33,34]. For the future integration of high-performance VLC with a high-resolution full-color display, the development of a long-wavelength micro-LED ranging from green to red light is essential [35,36]. Additionally, the low transmission loss of green light makes it an excellent choice for underwater communication and plastic optical fiber, making the development of high bandwidth green micro-LEDs crucial [37,38]. However, the existence of QCSE in the mainstream c-plane micro-LEDs induced by polarization related electric fields, makes it undesirable for VLC applications due to low modulation bandwidth. The polarization-related electric fields have been found to be reduced by growing the quantum wells on semipolar or nonpolar planes thereby providing room for bandwidth improvement [7,39,40]. The luminous efficiency of micro-LEDs is further affected by its small size, and performance degradation occurs due to serious etching issues. Despite having high current density for micro-LEDs, efficiency droop under high injection current density is also one of the major challenges that need to be addressed. As a result, it is difficult to achieve high efficiency and high modulation bandwidth at the same time, hence, there is still a large room for optimization when it comes to VLC performance. Due to the high efficiency and excellent bandwidth characteristics of GaN blue micro-LEDs, there has been a tremendous development of high-speed VLC systems over the past few decades. The development of long wavelength micro-LEDs ranging from yellow–green to red for VLC applications, however, requires considerable attention due to a few issues such as poor modulation bandwidth, efficiency droop, and instability at high temperatures. As a result, the modulation bandwidth of such long wavelength micro-LEDs must be improved in order to enable high-speed white light VLC systems consisting of communication and display. Some interesting reviews have been published on the overview of micro-LEDs for VLC, however, there is still a lack of reviews focusing on long wavelength high-speed micro-LEDs. In this review, we will first discuss the challenges being faced by micro-LEDs in achieving high modulation bandwidth for VLC applications. Recent progresses on micro-LEDs having emission wavelengths ranging from yellow–green to red wavelength will be reviewed and discussed in detail. In the final section, a summary and future perspective will be given.

## 2. Challenges

GaN/InGaN micro-LEDs have gained significant attention in recent years due to their potential for use in a wide range of applications. However, their performance is affected by several factors, including the quantum-confined Stark effect (QCSE), size-dependent effect, and droop effect. The combination of these factors significantly impacts the performance of micro-LEDs, such as modulation bandwidth, wavelength shift, full-width at half maximum (FWHM), light output power, and efficiency. Modulation bandwidth is essential for high-speed visible light communication. In addition, low wavelength shift and narrow FWHM can prevent crosstalk between different channels when applying wavelength division multiplexing, as well as improve the color accuracy of micro-LED displays. In this chapter, we will briefly introduce the phenomena and principles of these effects, as well as potential solutions to these challenges.

### 2.1. Quantum Confined Stark Effect

In this section, we examine the impact of the QCSE on the performance of GaN-based micro-LEDs. This effect, which arises from lattice mismatch within an InGaN/GaN quantum well, can cause a reduction in efficiency, a shift in wavelength, and a decrease in bandwidth [41,42]. It also leads to low carrier radiative recombination and low internal quantum efficiency (IQE) [43]. The polarization-induced electric field reduces the overlap of the electron-hole wave function, which in turn lowers the carrier recombination rate and thereby reduces the bandwidth of the micro-LED. As the indium content in the InGaN quantum well increases, the quantum-confined Stark effect (QCSE) becomes stronger, which is a phenomenon known as the “green gap”. This effect limits the development of green LEDs [44]. In previous research, we demonstrated that semipolar micro-LEDs could achieve higher bandwidths at lower current densities than micro-LEDs grown on polar c-plane GaN [45]. This is because semipolar micro-LEDs can improve the overlap of the electron-hole wave function due to lower QCSE, resulting in faster carrier recombination and therefore higher micro-LED bandwidth. However, at high current densities, the Coulomb screening effect can mitigate the QCSE, causing the change in band bending in a c-plane LED to be more significant than in a semipolar LED, leading to an increased bandwidth at a faster rate with current density. As a result, at high current density, the bandwidth of the polar LED may eventually approach that of the semipolar LED [40]. The −3 dB bandwidth of the micro-LED can be expressed as Equation (3) [46]:(3)$$f_{-3dB}=\frac{\sqrt{3}}{2\pi }\left(\frac{1}{\tau_{r}}+\frac{1}{\tau_{nr}}+\frac{1}{\tau_{RC}}\right)$$
where $\tau_{r}$, $\tau_{nr}$, and $\tau_{RC}$ are the carrier lifetimes of radiative recombination and non-radiative recombination and resonant cavity (RC) time constant, respectively.

In this simulation, systematic studies were numerically conducted to investigate the optoelectronic characteristics of InGaN-based green µ-LED using the commercial semiconductor device simulation tool, Crosslight. The settings of perfectly matched layers (PML) at the lateral and bottom surfaces of the µ-LED were expected to be the most important boundary conditions. The epitaxial structure can be composed of several layers, including a 2.5 μm undoped GaN layer, a 500 nm n-GaN layer with a doping concentration of 3 × 10^23^ m^−3^, five pairs of InGaN/GaN multiple quantum wells (MQWs) as the active region, followed by a 150 nm p-GaN layer with a doping concentration of 2 × 10^24^ m^−3^, and a 100 nm indium tin oxide (ITO) thin film as a current-spreading layer. The composition of the InGaN quantum wells is In_0.27_Ga_0.73_N(3.5 nm)/GaN(15 nm), In_0.305_Ga_0.695_N(3.5 nm)/GaN(15 nm), and In_0.315_Ga_0.685_N(3.5 nm)/GaN(15 nm) for c-plane, semipolar, and m-plane, respectively, resulting in an emission wavelength of 525 nm. The crystal orientation was set to c-plane (0001), semipolar, and m-plane.

Figure 1 shows the simulated band diagram and wavefunction overlap of c-plane (0001), semipolar ($20\bar{21}$), and m-plane ($10\bar{1}0$) micro-LEDs at 100 mA. Achieving long wavelength emission in InGaN-based micro-LEDs requires a high indium (In) content in the quantum well (QW) structure. However, the large lattice mismatch between indium nitride (InN) and gallium nitride (GaN) can cause defects and lattice strain during epitaxial growth. The performance of InGaN long wavelength micro-LEDs is limited by the quantum-confined Stark effect (QCSE), which reduces the external quantum efficiency (EQE) and causes a blue shift in the emission wavelength with increasing injected current. The QCSE is induced by a large built-in electron field, which arises from the polarization-induced electric field in the QW. The electric field separates the electrons and holes towards opposite sides of the QW, resulting in a lower radiative recombination rate and a corresponding Stark shift in excitonic absorption [47]. The EL intensity of InGaN-based LEDs increases with forward current injection, causing a blue shift in the emission peak due to the band-filling and screening effect of injected carriers [40]. The polarization field also bends the energy bands in the QW, reducing the transition energy between the first electron subband and the first hole subband, leading to a red-shift in EL emission. Applying a forward bias injects excess carriers into the QW and screens part of the polarization field, leading to a blue shift in the emission spectrum with increasing current density [48]. Due to the crystal orientation, as the current increases, the piezoelectric field tilts the band edge of the conduction band and valence band, leading to a decrease in wavefunction overlap, moreover, hindering the spontaneous radiative recombination rate, as expressed in Equation (4) [49]:(4)$$W_{spontaneous}\sim {\left|\langle u_{c}|u_{v}\rangle \right|}^{2}•{\left|\Gamma_{e\_hh}\right|}^{2}$$
where $|u_{c}\rangle$ and $|u_{v}\rangle$ are the Bloch functions for the conduction band and valence band, respectively, and $\left|\Gamma_{e\_hh}\right|$ is the envelope electron and hole wavefunction overlap.

The tilting of the band edge also causes an increase in the bandgap between the conduction band and valence band, resulting in a blue shift of the wavelength and a broadening of the full width at half maximum. The full width at half maximum (FWHM) of a GaN-based LED can be affected by the quality of the quantum well (QW), the well width, and indium fluctuations [50]. Since the well width is the same for all three types of micro-LEDs and there are no indium fluctuations in this simulation, the quality of the QW is the main factor. Because m-plane and semipolar micro-LEDs have less lattice mismatch than c-plane GaN micro-LEDs, they have better QW quality and narrower FWHM. The crystal orientation of the m-plane micro-LED is perpendicular to the direction of the piezoelectric field (c-axis), so it has a higher wavefunction overlap at high current densities. Figure 2 summarizes the optical performance of the simulated c-plane, semipolar, and m-plane micro-LEDs. As mentioned previously, due to the severe QCSE in the c-plane micro-LED, it has the largest wavelength shift of 5.5 nm and the broadest FWHM of 40 nm. The semipolar micro-LED has a wavelength shift of 3.5 nm and an FWHM of 35.5 nm. Finally, the m-plane micro-LED has a wavelength shift of 3 nm and an FWHM of 33.3 nm. The peak wavelength of the simulated c-plane, semipolar, and m-plane micro-LEDs is shown in Figure 3. Our simulation results are consistent with previous research [51,52,53,54,55] showing that, in general, m-plane and semipolar micro-LEDs perform better than c-plane micro-LEDs in terms of full width at half maximum (FWHM), bandwidth, and wavelength stability, due to the quantum-confined Stark effect (QCSE). However, with an increase in current density, the Coulomb screening effect screens the QCSE and the change of band bending in a c-plane LED is more significant than that in a semipolar LED, leading to an increased bandwidth at a faster rate with current density. As a result, at high current density, the bandwidth of the polar LED may eventually approach that of the m-plane and semipolar micro-LEDs [40].

Although semipolar and m-plane micro-LEDs have excellent performance, the cost and difficulty of mass production make them challenging. As a result, in recent years, researchers have been studying ways to improve the performance of c-plane micro-LEDs, such as using strain-released layer [41], superlattice [56], and staggered quantum well [49].

### 2.2. Size-Dependent Effect

In addition to the QCSE, self-heating can also impact the performance of micro-LEDs, including the bandwidth, wavelength shift, and efficiency, due to the shrinkage of the bandgap caused by self-heating. In general, the modulation bandwidth of LEDs is determined by the recombination lifetimes and RC time constant. However, for LEDs with sizes smaller than 100 μm × 100 μm, the radiative lifetime becomes the dominant factor in determining the modulation bandwidth because the geometric capacitance is small enough to prevent the RC time constant from becoming the dominant factor [46]. However, the current density is limited by the self-heating effect [57]. Smaller micro-LEDs can sustain higher current densities due to lower junction temperatures and reduced current crowding, resulting in faster radiative lifetimes. Therefore, high radiative recombination rates in micro-LEDs are crucial for implementing high-bandwidth VLC systems.

Increasing the injection current leads to a higher number of injected carriers (electrons and holes) in the active region, resulting in a higher carrier density and shorter differential carrier lifetime. The bandwidth will increase with higher current density [46] and eventually saturate due to the self-heating effect [57], which raises the junction temperature and hinders radiative recombination, causing an efficiency roll-off [58]. Fortunately, small-sized micro-LEDs can effectively reduce self-heating at high current densities, allowing for a wider operating range. At high current density, the radiative coefficient and Auger coefficient become the dominant factors influencing the bandwidth of the micro-LED.

Figure 4 indicates that the 50 μm × 6 micro-LED array fails when the current density surpasses 2500 A/cm^2^. On the other hand, the 30 μm × 8 micro-LED array exhibits a smaller rollover effect at a current density of up to 6500 A/cm^2^, indicating that it can handle a higher current density due to lower self-heating. Previous research by our team also revealed that the bandwidth improves with an increase in injection current density and a shorter carrier lifetime at different current density levels [41]. In addition to a high injection current, a small emitter diameter also contributes to uniform current spreading. As a result, a small-diameter LED can achieve a higher optical bandwidth even when the number of emitters is increased.

### 2.3. Droop Effect

The advantage of using micro-LEDs as a light source for VLC systems is their efficiency. However, as the injected current increases, the external (total) efficiency decreases, a phenomenon known as efficiency droop. The external quantum efficiency (*EQE*) of micro-LEDs is often expressed in terms of the internal quantum efficiency (*IQE*) and light extraction efficiency (*LEE*), as shown in Equation (5) [42]:(5)$$\eta_{EQE}=\eta_{IQE}\times \eta_{LEE}$$

Figure 5 shows the simulated internal quantum efficiency of c-plane (0001), semipolar ($20\bar{21}$), and m-plane ($10\bar{1}0$) micro-LEDs as a function of current. Figure 5 shows that the *IQE* drops as the injection current increases. The drop in the internal quantum efficiency at high current density is mainly due to Auger recombination and carrier leakage [42,59,60].

The impact of Auger recombination in internal quantum efficiency is usually expressed as Equation (6) [42]:(6)$$\eta_{IQE}=\frac{Bn^{2}}{\left(An+Bn^{2}+Cn^{3}\right)}$$
where *A*, *B*, *C*, and *n* are the Shockley–Read–Hall (SRH) recombination, spontaneous recombination, Auger recombination, and quantum well-carrier density, respectively. The low Auger recombination coefficient of semipolar and m-plane micro-LEDs [59,60] compared to the c-plane micro-LED [61] results in better *IQE* for both semipolar and m-plane micro-LEDs. The carrier leakage, which is caused by polarization due to the piezoelectric field, can also be observed in Figure 1. The flat GaN barrier energy in m-plane and semipolar micro-LEDs is caused by the absence of polarization fields, leading to a higher conduction band near the p-GaN side compared to the n-GaN side. This significantly reduces the electron leakage current and efficiency droop.

## 3. Progress for Micro-LED VLC in Yellow–Green to Red Emission Wavelength Range

As mentioned in the previous section, for micro-LEDs, the small luminous area limits their potential for long-distance communication due to limited light output power. Numerous approaches have been used to enhance output power by improving epitaxy quality and altering device structure; however, there is still a trade-off between output power and modulation bandwidth, and it is challenging to improve both characteristics simultaneously [62]. Designing series or parallel connections between micro-LED structures might alleviate this trade-off by significantly increasing the output power without compromising optical bandwidth. Micro-LED arrays in series emitting blue light have been demonstrated to have shown enhanced long-distance communication performance by exhibiting 11.74 Gbps/0.3 m and 1.61 Gbps/20 m [9]. While studies on series micro-LED array-based VLCs have only addressed blue light, no significant studies have taken into account the usage of series micro-LED arrays with various emission wavelengths to ultimately enhance VLC performance through the use of wavelength division multiplexing (WDM). In this regard, Zhu et al. recently demonstrated multicolor series connection micro-LED arrays and investigated optoelectronic and communication characteristics [15]. Figure 6a shows the optical micrograph images of the 3 × 3 series connection micro-LED arrays emitting green and yellow emission wavelengths. The green- and yellow-emitting micro-LED devices demonstrate a maximum −3 dB bandwidth of 348.1 and 93.9 MHz corresponding to the device structure of 40 μm and 20 μm, respectively, as shown in Figure 6b. At the desired BER of 3.8 × 10^−3^, the distributions of SNR, bit number, and power ratio on the subcarriers for green and yellow micro-LED arrays are produced as illustrated in Figure 6c,d. It is evident that the SNR for both devices at low frequencies is relatively small, which can be attributed to low frequency responses for bias-tee and amplifier. When the power loading is finished, the subcarrier power ratios then fluctuate around 1, but the overall power remains constant. The maximum achievable data transmission rates for the green and yellow micro-LED arrays are 4.39 and 0.82 Gbps, respectively. Figure 6e,f depict the power spectra of the green and yellow micro-LED arrays with the use of pre-equalization. The two devices exhibit a sever data rate loss due to the channel cross-talk effects induced by the spectral overlap.

In another study in 2021, Lin et al. demonstrated a green 2 × 2 micro-LED array with a high data transmission rate beyond 5 Gbit/s [63]. The schematic of the green 2 × 2 micro-LED array is shown in Figure 7a. The micro-LED was grown on a semipolar ($20\bar{21}$)-oriented GaN buffered GaN layer on ($22\bar{43}$)-oriented PSS. The unique structural design is advantageous for reducing the polarization-related electric field and suppressing the effect of QCSE. Therefore, as shown in Figure 7b, a very small wavelength shift of 7 nm was measured as the forward bias current increased from 5 mA to 100 mA, which retained good stability. Furthermore, there are higher polarization ratios compared to the same micro-LEDs which are grown on a c-plane PSS. By taking advantage of combining an ITO grating structure with Al-coated surface, the emission output power can be enhanced by 78% with increasing radiative recombination efficiency through simulations. The green micro-LED array has demonstrated a −3 dB and −6 dB bandwidths of 800 MHz and 1.02 GHz, respectively, however, due to the limited bandwidth of the photodiode, the bandwidths of the device have been reduced to 610 and 830 MHz, respectively, as shown in Figure 7c. The eye diagrams of the NRZ-OOK transmission at data rates of 1.5 Gbit/s are shown in Figure 7d. It is found that the eye remains clear and open; even the data rates at 1 Gbit/s and the maximum data rates exceed 1.5 Gbit/s for the NRZ-OOK format, showing a great deal of potential for optical wireless communication.

Future high-performance VLC with high-resolution full colour displays will necessitate the development of a long-wavelength micro-LED. Recently in 2022, Wei et al. utilized green InGaN quantum dots (QDs) as the active region of micro-LED to alleviate the limitation of bandwidth due to the long radiative recombination carrier lifetime in InGaN QW [64]. A green micro-LED with five-layer of InGaN QDs above the 10-layer superlattices is shown in Figure 8a. The SiO2 passivation layer formed by plasma-enhanced chemical vapor deposition (PECVD) is used to reduce the leakage current. The carrier lifetime of the subnanometer-height QD layer could be measured by the time-resolved photoluminescence (TRPL), and its value was possibly as short as 560 ps due to a weak QCSE and small QD sizes. Given their short carrier lifetime, QDs offer a lot of potential for use in high-speed VLC applications. The normalized frequency responses of 50- and 75 μm diameter QD green micro-LED are shown in Figure 8b. As mentioned above, the 50 μm QD green micro-LED has a wider modulation bandwidth than the 75 μm one, while the extracted modulation bandwidth increases with the increase in the injected current density. The maximum bandwidth of 1.22 GHz and 1.14 GHz are measured, respectively. We can implement a high-capacity GHz VLC system with a low current density thanks to this wide −3 dB modulation bandwidth. The communication performances of real-time NRZ-OOK experiments are shown in Figure 8c,d. With setting a higher data rate, the BER is becoming worse and reaching the forward error correction limit. On the other hand, the BER performance shows improvement in increasing injected current density. It is obvious to observe that the clarity of eye diagrams deteriorates gradually with the increasing data rate as shown in Figure 4d. The eye diagram at the data rate of 2 Gbit/s is almost closed; however, by satisfying the forward error correction limit of 3.8 × 10^−3^, the highest data rate of 2.1 Gbit/s is achieved at the current density of 384.81 A/cm^2^. The outcomes based on green QD micro-LEDs offer enormous potential for VLC applications for the next generation.

During the past few years, structural optimization for modulation bandwidths has been reported frequently. In 2022, Huang et al. demonstrated a high −3 dB bandwidth yellow–green InGaN-based micro-LED using nanoporous distributed Bragg reflectors to increase light extraction efficiency [41]. The yellow–green micro-LED was grown on a polar c-plane GaN epitaxial wafer and with PSS at the bottom of the device structure as shown in Figure 9a. Three pairs of InGaN/GaN multiple QWs (MQWs) acted as the active region in the structure. The nanoporous (NP) DBR was introduced into the micro-LED structure, and the NP-DBR was designed to reflect the light at wavelengths of 500 to 600 nm; therefore, improving the yellow–green light extraction efficiency. Moreover, the NP-DBR can also serve as a strain-released layer for the active layer to mitigate the effect of QCSE, leading to enhance EQE. Besides, the ALD technology was used to suppress the influence of the sidewall defects, resulting in reducing the leakage current. The electroluminescence (EL) emission wavelength was shown in Figure 9b. When the injected current density rose from 13 A/cm^2^ to 2500 A/cm^2^, the peak wavelength blueshifted from 574 nm to 547 nm; meanwhile the emission wavelength exhibited a steady wavelength of 547 nm at high current density. The small wavelength shift and stability are regarded as excellent performance compared to conventional yellow–green micro-LEDs. The frequency response at −3 dB through different diameters and numbers is summarized in Figure 9c. The −3 dB bandwidth is found to be proportional to the current density, and only relevant to the emitter diameter instead of the number of emitters. The larger diameter of the emitter, the more serious the self-heating effect of the micro-LED array is, leading to the earlier roll-off point of the light, and the bandwidth is inversely proportional to the emitter diameter. Therefore, the maximum bandwidth of 442 MHz was observed at 2500 A/cm^2^ for a 30 μm diameter micro-LED array. The measured eye diagrams for the micro-LED array at an injected current density of 2500 A/cm^2^ are shown in Figure 9d. The eye is clear and open at 200 Mbit/s, and gradually becomes smaller when the bit rate is up to 800 Mbit/s, where the bit error rate is satisfying the forward error correction limit. The results reveal that VLC can be implemented via a yellow–green micro-LED.

The application of high-efficiency GaN-based micro-LEDs for VLC has been demonstrated in a number of recent experiments employing modulation techniques such as OFDM or NRZ-OOK, demonstrating Gbps data transmission rates. The extraordinarily high bandwidth at a low current density of nonpolar GaN-based micro-LEDs, in particular, makes them especially promising for VLC applications. AlGaInP LEDs, on the other hand, do not have internal electric fields associated with polarization, which makes them a potential option for data transmission in micro-LED geometries. In 2020, Carreira et al. demonstrated red-emitting aluminum gallium indium phosphide micro-LEDs via micro-transfer printing for VLC applications [65]. The micro-LED platelets were printed on glass and diamond substrates, with an emission wavelength of 630 nm, and have an epitaxial structure that is similar to typical AlGaInP epistructures produced on GaAs. A photograph showing a 2 × 2 micro-LED array (P) was printed on glass substrate along with single micro-LEDs (S1) driven at 4.3 A/cm^2^ is shown in Figure 10a. Polydimethylsiloxane (PDMS) stamps were used to imprint the micro-LED platelets onto both substrates after they had been picked up from the temporary sapphire substrate. After the printing process is completed, the device was encapsulated using Parylene-C (Pa-c) layer, having a thickness of 4 um. The modulation performance for the single micro-LED on different substrates is demonstrated in Figure 10b illustrating the −6 dB electrical modulation bandwidth with respect to the applied current density. As shown in the figure, both micro-LEDs on diamond and glass substrates exhibit similar modulation characteristics below 100 A/cm^2^. However, the micro-LED on a diamond substrate exhibited a higher modulation bandwidth reaching a maximum value of 170 MHz at 1000 A/cm^2^ while the micro-LED on the glass substrate can reach only up to 85 MHz at 431 A/cm^2^. Figure 10c shows the signal-to-noise ratio (SNR) and the number of allocated bits as a function of frequency for the single micro-LED on both substrates. The SNR for the micro-LED on glass suffers a severe decline in frequency and can only sustain bit loading up to 656 MHz. While the micro-LED on diamond demonstrates a steady decline in SNR with frequency and bit loading up to 1245 MHz. The bit-error-ratio (BER) of a single micro-LED and in-parallel micro-LED array on both substrates as a function of data rate is shown in Figure 10d. The single micro-LED on glass and diamond can achieve error-free data rates of 2557 and 5014 Mbps, respectively, corresponding to a BER below the FEC threshold. For the in-parallel micro-LED array, the error-free data rates are 3034 and 6596 Mbps for glass and diamond substrates, respectively.

Although AlGaInP-based red micro-LEDs demonstrated higher VLC performance, they had serious surface recombination problems that cause instability at high temperatures and size-dependent efficiency droop [66,67,68,69,70,71,72]. As a result, it is desirable to develop monolithic RGB micro-LED technology using InGaN-based red micro-LEDs. Blue and green micro-LEDs based on InGaN have been shown to perform admirably in VLC applications, however, red micro-LEDs grown on c-planes still suffer from a substantial efficiency drop due to the high indium content requirement for red light emission. High In concentration will cause QCSE and impede radiative recombination due to the significant lattice mismatch between InN and GaN limiting the modulation bandwidth. Although the QCSE of the LEDs can be decreased by growing them on semipolar substrates, InGaN red micro-LEDs grown on c-plane substrates are currently the best option due to their compliance with existing manufacturing techniques. Recently in 2022, Huang et al. demonstrated red-emitting InGaN-based micro-LED with high efficiency and high modulation bandwidth by incorporating superlattice structure and distributed Bragg reflector [56]. The epitaxial structure of the LED comprised blue InGaN single-QW (SQW) and red InGaN double-QWs (DQWs) as the active region. The epitaxial structure was grown on a c-plane patterned sapphire substrate (PSS) using metal-organic vapor-phase epitaxy (MOVPE). As shown in Figure 11a, there are three key technologies in the design of the device. First, the use of a superlattice (SL) structure, as a buffer layer, was inserted to release the stress of QW of the red micro-LED, thereby improving the epitaxial quality and suppressing the effect of QCSE. Second, the atomic layer deposition (ALD) passivation technique was introduced [56]. The ALD layer not only helped to passivate the surface defects caused by dry etching but reduced the leakage current on the sidewall of the devices [56]. Third, a distributed Bragg reflector (DBR) was designed at the bottom of the packaged device to enhance the light extraction efficiency. The high external quantum efficiency (EQE) of the red InGaN micro-LED is shown in Figure 11b. The 25 μm diameter red micro-LED exhibited a maximum EQE value of 5.02% with an efficiency drop of only 9.1% when the injection current density is up to 400 A/cm^2^. The frequency response of the 6 × 25 μm diameter InGaN red micro-LED array at different injection current densities is shown in Figure 11c. The frequency response at −3 dB is found to increase with the injection current density increasing. The 25 μm diameter LED has the highest frequency response of 271 MHz at the injection current density of 2000 A/cm^2^. The data transmission characteristics of the red micro-LEDs were also analyzed by an NRZ-OOK modulation format. Figure 11d illustrates the eye diagrams for 25 μm diameter red micro-LED at 200 Mbit/s, 300 Mbit/s, and 390 Mbit/s. The eye diagram is clear and open at 200 Mbit/s, while the eye is beginning to close at 300 Mbit/s. At 390 Mbit/s, the eye region is not apparent and is virtually closed. The eye diagram shows the potential application of the red micro-LED at data rates of the order of 300 Mbit/s. The process indicates that InGaN red micro-LEDs are promising for VLC and full-color microdisplay applications. Even though InGaN red micro-LEDs exhibited superior VLC performance, there is still room for improvement and optimization for future-generation optoelectronics.

A benchmark for the previously reported modulation bandwidth of micro-LED arrays emitting yellow–green to red wavelength for VLC performance is given in Table 1. The studies on blue high-speed micro-LEDs are far more prevalent than those on other wavelengths since blue LEDs are the most advanced of all visible LEDs and are also the basis of white LEDs. While red and yellow micro-LEDs generally have bandwidths below 500 MHz due to high In content, green micro-LEDs typically have bandwidths below 1 GHz. By employing an array configuration, only a few of the micro-LEDs with green emission wavelengths can approach 1 GHz since they can achieve high injection current densities that can mitigate the polarization effects. When compared to polar devices, nonpolar micro-LEDs and micro-LEDs with micro-structures have been shown to achieve the highest modulation bandwidths, especially for green emission wavelength. Other wavelength MQW micro-LEDs, such as those that emit yellow or red light, may not exhibit obvious bandwidth performance, which may be associated with the screening effect under high current density. It is important to understand that the communication speed is not just completely reliant on the device’s response time but also on SNR.

## 4. Summary and Perspective

The advancement of micro-LED technology has increased the perception of this state of the art due to its superior display and communication capabilities. It has been demonstrated that micro-LEDs can achieve high modulation bandwidth due to low RC time constant and their ability to attain higher current density. However, it is challenging for micro-LED to simultaneously attain high efficiency and high speed due to QCSE, size dependence factor, and efficiency droop effect. In this review, we present a comprehensive review of the state of-the-art regarding high-speed micro-LEDs having yellow–green to red emission wavelengths. We have also presented various challenges posed by micro-LEDs such as QCSE, size-dependency, and droop effect that is responsible for controlling modulation bandwidths. Different approaches in terms of structural optimization for modulation bandwidth enhancement have also been presented and analyzed. 

Using blue micro-LEDs with quantum dots for color conversion is a potential solution that has been extensively studied. By using blue micro-LEDs and coating them with quantum dots of the desired color, it is possible to achieve a broader range of colors, including red, green, and yellow. This approach is known as a color conversion technique [82,83]. One of the main advantages of this technique is that it can provide highly efficient and stable color conversion with narrow FWHM, which is essential for display applications [84]. However, there are some challenges to using this approach. One of the main challenges is that quantum dots have limited stability and can degrade over time. Another challenge is that the process of depositing the quantum dots onto the micro-LED surface can be complex and may require careful optimization [83,85].

Despite the fact that micro-LEDs and their array structure have demonstrated superior VLC performance, there is still room for further advancement. Because of their small size, micro-LEDs can easily obtain high current densities, which is beneficial for modulation speed but causes some issues with luminous efficiency. As a result, it is important to develop micro-LEDs having high efficiency and high modulation bandwidth at the same time, which will be a future direction of research. Some of the potential solutions include enhancing the efficiency droop, minimizing etching damage, and achieving high bandwidth at a low current. Along with enhancing device performance, we must develop advanced modulation formats appropriate for VLC applications, which will help increase the modulation bandwidth for micro-LEDs and their array structure.

## Figures and Tables

**Figure 1 micromachines-14-00478-f001:**
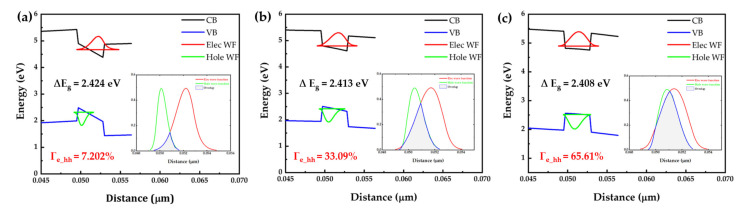
The simulated band diagram and wavefunction overlap of (**a**) c-plane (0001), (**b**) semipolar ($20\bar{21}$), and (**c**) m-plane ($10\bar{1}0$) micro-LED at 100 mA. (CB: Conduction band; VB: Valence band; Elec WF: Electron wavefunction; Hole WF: Hole wavefunction;$\Delta E_{g}$: Bandgap; $\Gamma_{e\_hh}$: The envelope electron and hole wavefunction overlap).

**Figure 2 micromachines-14-00478-f002:**
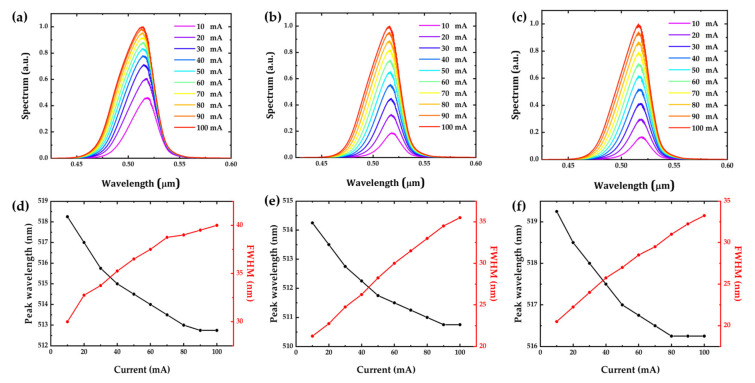
The emission spectrum of the (**a**) c-plane (0001), (**b**) semipolar ($20\bar{21}$), and (**c**) m-plane ($10\bar{1}0$) micro-LED, and the wavelength shift and FWHM as a function of current of the (**d**) c-plane (0001), (**e**) semipolar ($20\bar{21}$), and (**f**) m-plane ($10\bar{1}0$) micro-LED.

**Figure 3 micromachines-14-00478-f003:**
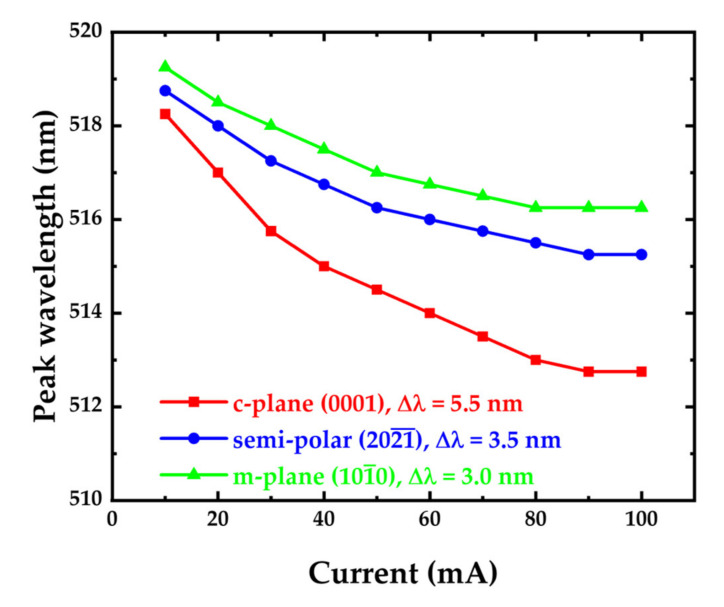
The peak wavelength of the simulated c-plane (0001), semipolar ($20\bar{21}$), and m-plane ($10\bar{1}0$) micro-LEDs as a function of current.

**Figure 4 micromachines-14-00478-f004:**
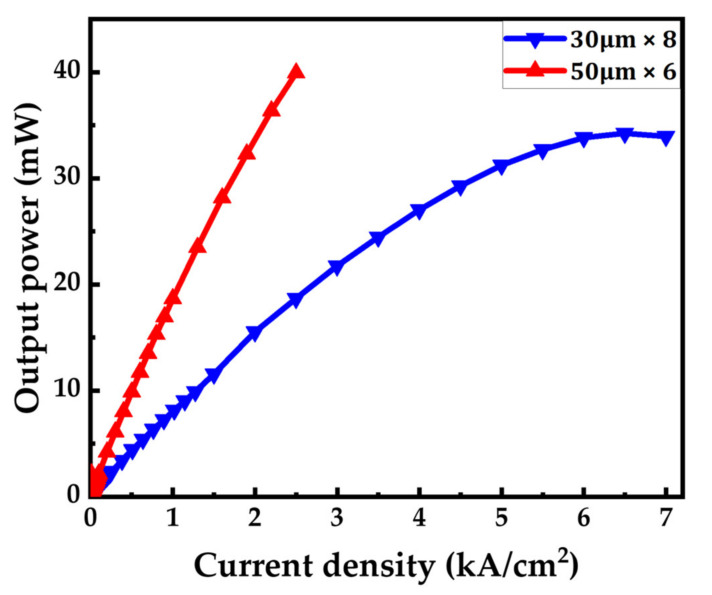
The current density versus optical power of 30 μm × 8 and 50 μm × 6 micro-LED arrays.

**Figure 5 micromachines-14-00478-f005:**
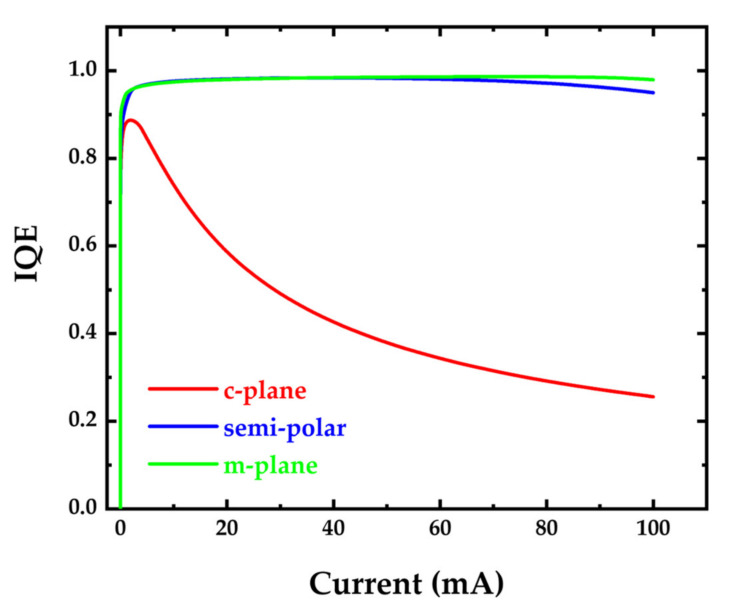
The simulated internal quantum efficiency of c-plane (0001), semipolar ($20\bar{21}$), and m-plane ($10\bar{1}0$) micro-LEDs as a function of current.

**Figure 6 micromachines-14-00478-f006:**
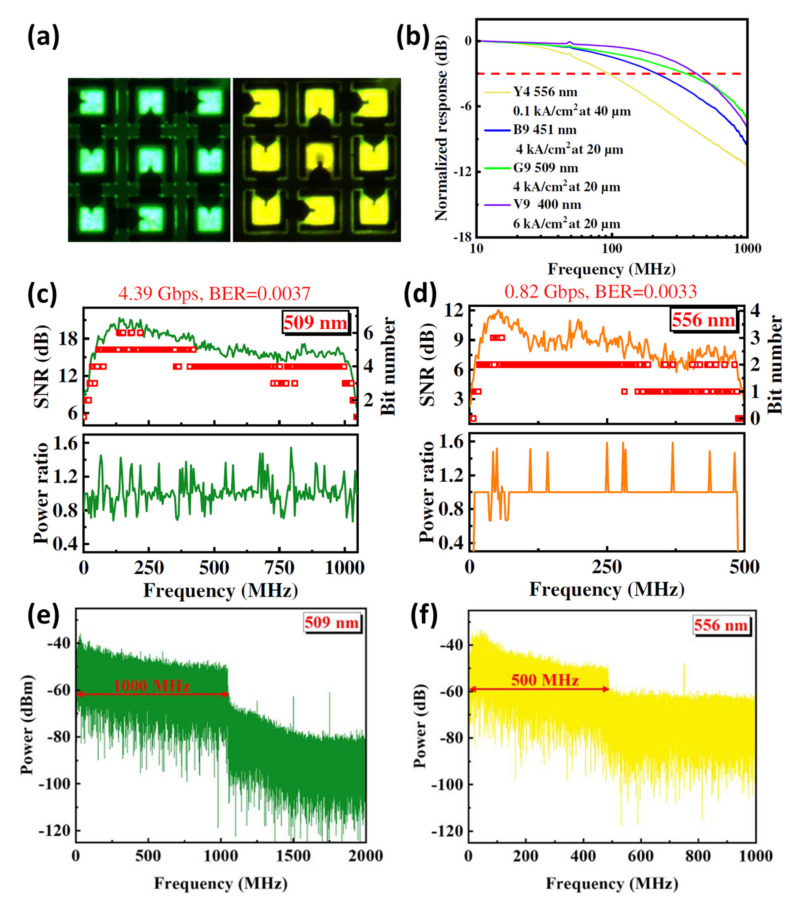
(**a**) Optical micrographs for the green and yellow micro-LED arrays. (**b**) Modulation bandwidths of the micro-LED arrays. (**c**,**d**) SNR and bit number (**e**,**f**) Power spectra for the green and yellow micro-LED arrays, respectively. Ref. [15] Figures reproduced with permission from Optica Publishing Group.

**Figure 7 micromachines-14-00478-f007:**
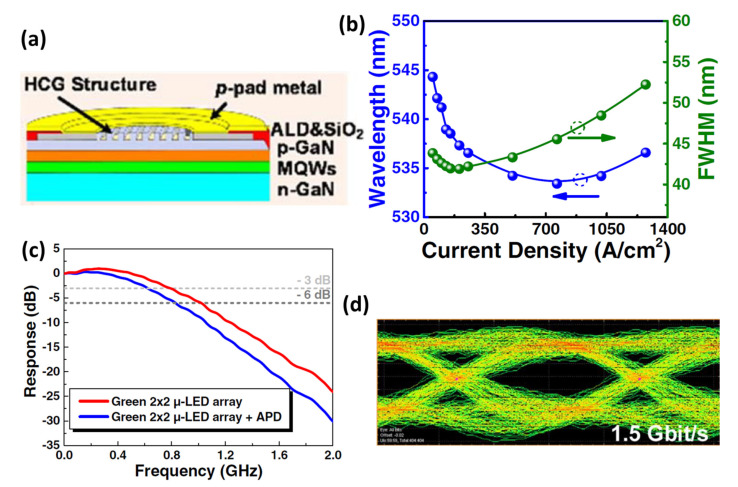
(**a**) Schematic diagram of micro-LED device structure within nanostructured grating patterns on top of the structure. (**b**) Peak wavelength and FWHM of the EL spectra. (**c**) Modulation bandwidth of the micro-LED array. (**d**) Eye diagram of the NRZ-OOK transmission at 1.5 Gbit/s. Ref. [63] Figures reproduced with permission from Optica Publishing Group.

**Figure 8 micromachines-14-00478-f008:**
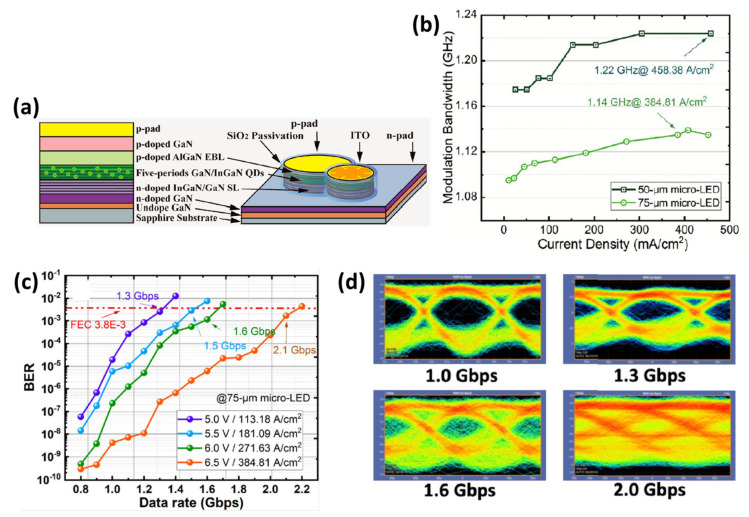
(**a**) Schematic diagram of epitaxial and single-pixel QD micro-LED structure. (**b**) Frequency responses of VLC systems based on 50 and 75-μm green QD micro-LEDs. (**c**) BER performances and (**d**) Eye diagrams of real-time NRZ-OOK measurement. Ref. [64] Figures reproduced with permission from the American Chemical Soceity (ACS).

**Figure 9 micromachines-14-00478-f009:**
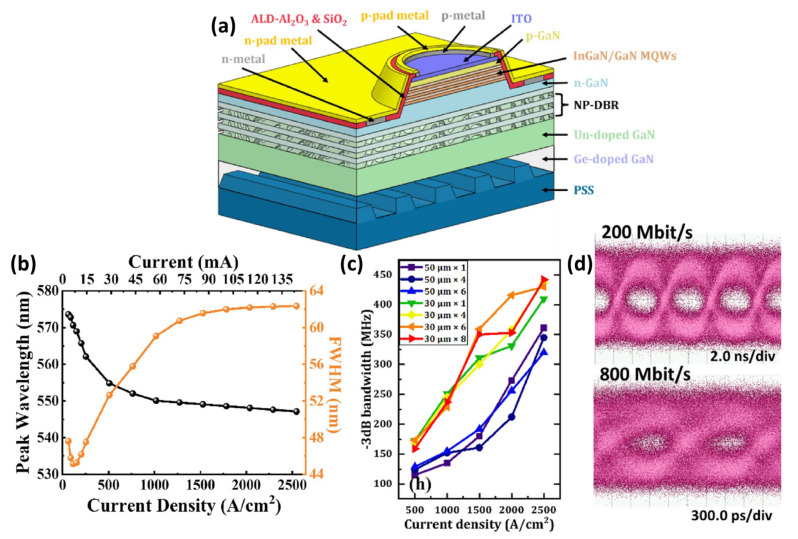
(**a**) Schematic diagram of yellow–green micro-LED structures. (**b**) Electroluminescence wavelength shift and FWHM as a function of current density (**c**) The −3 dB bandwidth through different diameters and numbers of micro-LED arrays. (**d**) Eye diagrams for 30 μm × 8 micro-LED array at 200 Mbit/s and 800 Mbit/s. Ref. [41] Figures reproduced with permission from Optica Publishing Group.

**Figure 10 micromachines-14-00478-f010:**
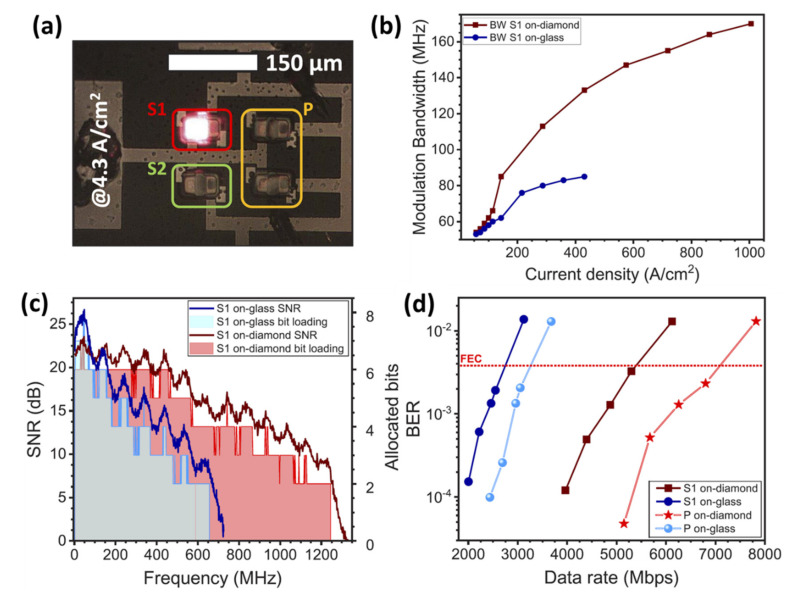
(**a**) Single micro-LED (S1) and in-parallel micro-LED array (P) on glass driven at 4.3 A/cm^2^. (**b**) Modulation bandwidth vs. current density for the single micro-LED on both substrates. (**c**) SNR and number of allocated bits vs. frequency for single micro-LEDs on both substrates. (**d**) BER vs. data rate for single and in-parallel micro-LEDs on both substrates. Ref. [65] Figures reproduced with permission from Optica Publishing Group.

**Figure 11 micromachines-14-00478-f011:**
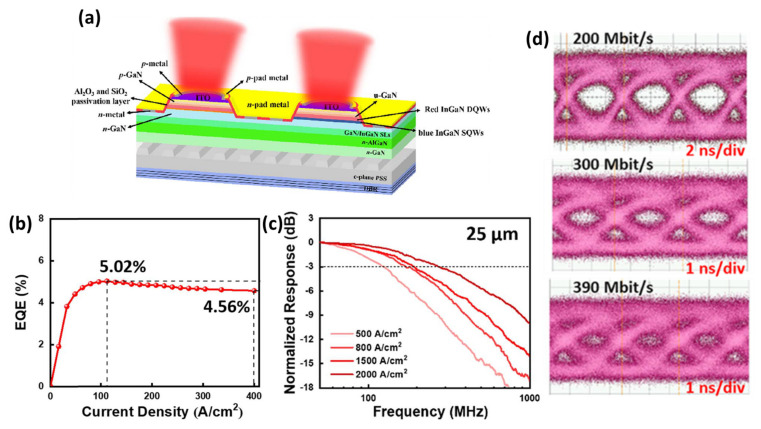
(**a**) Schematic diagram of c-plane InGaN red micro-LED structure. (**b**) EQE as a function of current densities for 25 μm sized micro-LED. (**c**) Frequency response measurement of 25 μm sized micro-LED. (**d**) Eye diagrams for 25 μm sized micro-LED. Ref. [56] Figures reproduced with permission from Optica Publishing Group.

**Table 1 micromachines-14-00478-t001:** Modulation characteristics comparison between Micro-LEDs and mini-LEDs emitting yellow–green to red wavelengths.

Color	Structure	Bandwidth (MHz)	Modulation Format	Data Rate (Gbps)	BER	Year	Ref.
Green		340	NRZ-OOK	0.4	<FEC	2018	Yeh et al. [73]
Green	75 um	340	QAM-OFDM	2.16	<FEC	2018	Chen et al. [74]
Green	100 um	144	QAM-OFDM	0.62	Error free	2019	Carreira et al. [75]
Green	50 um	756	NRZ-OOK	1.5	<FEC	2020	Chen et al. [45]
Green Yellow Amber	330 × 330 um^2^	540 350 150	QAM-OFDM	4.22 3.72 0.336	1.8 × 10^−3^ 1.4 × 10^−3^ 3.6 × 10^−3^	2020	Hagger et al. [76]
Green	2 × 3 array	1060	QAM-OFDM	3.129	2.01 × 10^−3^	2021	Chang et al. [77]
Green	2 × 2 array	800	NRZ-OOK QAM-OFDM	1.5 5.02	3.3 × 10^−3^	2021	Lin et al. [63]
Green	2 × 3 array	1102	QAM-OFDM	4.343	2.47 × 10^−3^	2021	Chang et al. [78]
Green Yellow	3 × 3 array	1050 500	QAM-OFDM	4.39 0.82	3.7 × 10^−3^ 3.3 × 10^−3^	2022	Zhu et al. [15]
Yellow		238	NRZ-OOK	650 Mbps	1 × 10^−6^	2016	Luo et al. [79]
Yellow		300	QAM-DMT	1.25	3.8 × 10^−3^	2018	Zhu et al. [80]
Yellow		260	OFDM	1.25		2020	Milovančev et al. [81]
Yellow	30 um	442	NRZ-OOK	800 Mbps		2022	Huang et al. [41]
Red	2 × 1 array	170	DCO-OFDM	7093 Mbps	3.8 × 10^−3^	2020	Carreira et al. [65]
Red	6 × 25 um	271	NRZ-OOK	350 Mbps	2.6 × 10^−3^	2022	Huang et al. [56]

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
