# Peer review of "Recent Advances in Micro-LEDs Having Yellow–Green to Red Emission Wavelengths for Visible Light Communications"

_micromachines, 2023, doi:10.3390/mi14020478_

Round 1
Reviewer 1 Report
Visible light communication (VLC) has an important application prospect in supporting high-speed Internet connection and wireless communication. As a VLC communication light source, micro-LED is used in VLC due to its high bandwidth, wide spectral band and other characteristics and potential in lighting and communication. However, due to the impact of low output optical power efficiency, bandwidth and other factors, it has not yet entered a wide range of practical applications.
In this review, the authors summarize the current research progress of VLC LED devices, discuss in detail the current status and significant bandwidth-enhancing progress of mid-wavelength and long-wavelength visible light micro-LEDs for VLC, and the impact of micro-LED applications in VLC. factors, including quantum confinement stark effects, size-dependent effects, and droop effects. This discusses the challenges and future prospects of micro-LEDs for VLC.
This review is comprehensive and in-depth, and is a good reference for researchers working on micro-LEDs and VLCs. Suggested posting.
Author Response
We thank the reviewer for the positive comments.
Reviewer 2 Report
The article addresses a review about recent advances in micro-LEDs having yellow green to red 2 emission wavelengths for visible light communications.
General comment: The article needs some major improvements and at the moment to have opportunity for the second round of the review.
1- The abstract should be modified. There is not enough information about the methodology, proposed works, conclusion y comparison with other works in this part. Also, the abstract can be rewritten longer but with more details and some numerical results.
I suggest you structure your abstract as presented in https://www.principiae.be/pdfs/UGent-X-003-slideshow.pdf
2- The introduction has been written very short and it should be extended to new published papers for recent years. In the introduction should be expressed the better state-of-art of new methods. The new references will also be examined in this part. I would like to see the articles for last and this year in this section.
3- There are many paragraphs in the introduction, where more of them can be merged to have better structure.
4- Since the article is a review, it needs a better state-of-the-art, where there are many works related to this topic during last years, where had not been mentioned in this article.
5- I cannot see the details of the methods in this article. The methods were explained very short without preparing enough explanation.
6- Figures 1 and 2 should be explained in a better way. There are some details, which have not been explained in the article.
7- Simulation conditions are not well discussed. The approaches were illustrated only on some specific simulations, which is not enough to draw a complete and accurate conclusion about the methods.
8- The evaluations in not enough. The comparisons have been shown in some specific cases, where in my point of view there is no good comparison in the methods. There is no appropriate comparison with other previous works.
9- As a review paper, the presented methods seem separated and it needs much better connections between introduced algorithms.
10- Please, do not forget that the clarity and the good structure of an article are important factors in the review decision. Please read the paper carefully (again) and correct it in English.
Author Response
The article addresses a review about recent advances in micro-LEDs having yellow green to red 2 emission wavelengths for visible light communications.
General comment: The article needs some major improvements and at the moment to have opportunity for the second round of the review.
We thank the reviewer for taking time to review our manuscript.
1- The abstract should be modified. There is not enough information about the methodology, proposed works, conclusion y comparison with other works in this part. Also, the abstract can be rewritten longer but with more details and some numerical results.
I suggest you structure your abstract as presented in https://www.principiae.be/pdfs/UGent-X-003-slideshow.pdf
Response:
Thank you very much for the comments. The abstract have been modified accordingly and more contents have been added to this section. The proposed simulation have been added in the corresponding sections and the conclusion have been modified accordingly.
2- The introduction has been written very short and it should be extended to new published papers for recent years. In the introduction should be expressed the better state-of-art of new methods. The new references will also be examined in this part. I would like to see the articles for last and this year in this section.
Response:
Thank you very much for the comments. We have made necessary changes to the manuscript.
3- There are many paragraphs in the introduction, where more of them can be merged to have better structure.
Response:
Thank you very much for the comments. We have made necessary changes to the manuscript.
4- Since the article is a review, it needs a better state-of-the-art, where there are many works related to this topic during last years, where had not been mentioned in this article.
Response:
Thank you very much for the comments. We have made necessary changes to the manuscript.
5- I cannot see the details of the methods in this article. The methods were explained very short without preparing enough explanation.
Response:
Thank you very much for the comments. In this simulation, systematic studies were numerically conducted to investigate the optoelectronic characteristics of InGaN-based green µ-LED using the commercial semiconductor device simulation tool Crosslight. The settings of perfectly matched layers (PML) at the lateral and bottom surfaces of the µ-LED were expected to be the most important boundary conditions. The epitaxial structure can be composed of several layers, including a 2.5 μm undoped GaN layer, a 500 nm n-GaN layer with a doping concentration of 3×1023 m-3, five pairs of InGaN/GaN multiple quantum wells (MQWs) as the active region, followed by a 150 nm p-GaN layer with a doping concentration of 2×1024 m-3, and a 100 nm indium tin oxide (ITO) thin film as a current-spreading layer. The composition of the InGaN quantum wells is In0.27Ga0.73N(3.5 nm)/GaN(15 nm), In0.305Ga0.695N(3.5 nm)/GaN(15 nm), and In0.315Ga0.685N(3.5 nm)/GaN(15 nm) for c-plane, semipolar, and m-plane, respectively, resulting in an emission wavelength of 525 nm. The crystal orientation was set to c-plane (0001), semipolar, and m-plane. The above description has been added on page 4.
6- Figures 1 and 2 should be explained in a better way. There are some details, which have not been explained in the article.
Response:
Thank you very much for the comments. We have added the following description to further explain Figures 1 and 2 on page 4 and 5, as per your suggestion.: “To achieve long wavelength emission, a high In-content InGaN/GaN quantum well (QW) structure is required, but the large lattice mismatch between InN and GaN makes the epitaxial process challenging, resulting in a high number of defects and lattice strain in high-In-content QWs. The performance of InGaN long wavelength micro-LEDs is limited by the quantum-confined Stark effect (QCSE), which causes a low external quantum efficiency (EQE) and a blue shift of emission wavelength as injected current increases. When an electric field is applied perpendicularly to the QW layer, electrons and holes separate towards opposite sides, causing a lower radiative recombination rate and a corresponding Stark shift in excitonic absorption [47]. The electro-luminescence (EL) intensities of InGaN-based LEDs increased with forward current injection, leading to a blue shift of the emission peak due to the band-filling and screening effect of injected carriers [40]. The QCSE in InGaN-based micro-LEDs arises from a large polarization-induced electric field, also called a built-in electron field, which bends the energy bands in the QW, reducing the transition energy from the first electron subband to the first hole subband and causing a red-shift of EL emission. Applying a forward bias injects excess carriers into the QW and screens part of the polarization field, leading to a blue shift in the spectrum with increasing current density.”
7- Simulation conditions are not well discussed. The approaches were illustrated only on some specific simulations, which is not enough to draw a complete and accurate conclusion about the methods.
Response:
Thank you very much for the comments. In this simulation, systematic studies were numerically conducted to investigate the optoelectronic characteristics of InGaN-based green µ-LED using the commercial semiconductor device simulation tool Crosslight. The settings of perfectly matched layers (PML) at the lateral and bottom surfaces of the µ-LED were expected to be the most important boundary conditions. The epitaxial structure can be composed of several layers, including a 2.5 μm undoped GaN layer, a 500 nm n-GaN layer with a doping concentration of 3×1023 m-3, five pairs of InGaN/GaN multiple quantum wells (MQWs) as the active region, followed by a 150 nm p-GaN layer with a doping concentration of 2×1024 m-3, and a 100 nm indium tin oxide (ITO) thin film as a current-spreading layer. The composition of the InGaN quantum wells is In0.27Ga0.73N(3.5 nm)/GaN(15 nm), In0.305Ga0.695N(3.5 nm)/GaN(15 nm), and In0.315Ga0.685N(3.5 nm)/GaN(15 nm) for c-plane, semipolar, and m-plane, respectively, resulting in an emission wavelength of 525 nm. The crystal orientation was set to c-plane (0001), semipolar, and m-plane. The above description has been added on page 4.
8- The evaluations in not enough. The comparisons have been shown in some specific cases, where in my point of view there is no good comparison in the methods. There is no appropriate comparison with other previous works.
Response:
Thank you very much for the comments. Our simulation results are consistent with previous research [51-55] showing that, in general, m-plane and semipolar micro-LEDs perform better than c-plane micro-LEDs in terms of full width at half maximum (FWHM), bandwidth, and wavelength stability, due to the quantum-confined Stark effect (QCSE). However, with an increase in cur-rent density, the Coulomb screening effect screens the QCSE and the change of band bending in a c-plane LED is more significant than that in a semipolar LED, leading to an increased bandwidth at a faster rate with current density. As a result, at high current density, the bandwidth of the polar LED may eventually approach that of the m-plane and semipolar micro-LEDs [40]. The above description has been added on page 7.
9- As a review paper, the presented methods seem separated and it needs much better connections between introduced algorithms.
Response:
Thank you very much for the comments. We have made necessary changes in the manuscript.
10- Please, do not forget that the clarity and the good structure of an article are important factors in the review decision. Please read the paper carefully (again) and correct it in English.
Response:
Thank you very much for the comments. We have made necessary changes in the manuscript.

Reviewer 3 Report
This manuscript describes the challenges and review the progress in long-wavelength InGaN microLEDs. The manuscript requires mandatory revision before considering for acceptance. There are a lot of improvements to be made in the challenge section, as well as a better literature search. Below are questions and comments that the authors need to consider for revision.
1. The x-axis of Fig 2a is incorrect. Please revise it. Also, the authors should normalize the y-axes in all figures, otherwise some of them are difficult for comparison. The FWHM is also affected by the material quality of the active region. Why is the FWHM of c-plane is greater than semi-polar and non-polar? The wavelength shift is smaller than reports in the literature, can the authors explain this? The authors should specify if this is experimental or simulation results, since high indium growths on m-plane are challenging.
2. The authors point out long-wavelength microLEDs are attractive because of the use in displays. However, it seems like blue microLEDs plus quantum dots for color conversion could be a potential solution to address most issues discussed in the manuscript, such as wavelength shift and wide FWHM. Can the authors discuss this alternative?
3. The size effect also introduces sidewall defects that could increase the RC time constant, yet the authors did not mention this at all. Is this not a potential issue for microLED VLC? It seems like the 50 um devices give better LOP than the 30 um devices, this is true even if LOP is normalized by the area. Smaller microLEDs should give better LOP per area than the larger counterpart, why is it not the case here?
4. The authors claimed AlGaInP red microLEDs are not suitable for VLC due to the size effect. What about their optical properties compared with InGaN red LEDs? The authors cited a AlGaInP microLEDs with sidewall treatments, so the mitigation of the size effect in AlGaInP microLEDs is possible. Also, references for ALD passivation and the development of InGaN red LEDs are lacking.
I would recommend to proofread the manuscript for language use. Some terms and sentences like "... is very relatively rare" and "When the indium content in the InGaN quantum well increases, the QCSE becomes stronger, a phenomenon known as the "green gap" that limits the development of green LEDs" should be corrected to improve readability before resubmission.
Author Response
This manuscript describes the challenges and review the progress in long-wavelength InGaN microLEDs. The manuscript requires mandatory revision before considering for acceptance. There are a lot of improvements to be made in the challenge section, as well as a better literature search. Below are questions and comments that the authors need to consider for revision.
We thank the reviewer for taking time to review our manuscript.
- The x-axis of Fig 2a is incorrect. Please revise it. Also, the authors should normalize the y-axes in all figures, otherwise some of them are difficult for comparison. The FWHM is also affected by the material quality of the active region. Why is the FWHM of c-plane is greater than semi-polar and non-polar? The wavelength shift is smaller than reports in the literature, can the authors explain this? The authors should specify if this is experimental or simulation results, since high indium growths on m-plane are challenging.
Response:
Thank you very much for the comments. We have corrected the x-axis of Fig 2a to c and normalized the y-axis as per your suggestion.
The full width at half maximum (FWHM) of a GaN-based LED can be affected by the quality of the quantum well (QW), the well width, and indium fluctuations [37]. Since the well width is the same for all three types of micro-LEDs and there are no indium fluctuations in this simulation, the quality of the QW is the main factor. Because m-plane and semipolar micro-LEDs have less lattice mismatch than c-plane GaN micro-LEDs, they have better QW quality and narrower FWHM. The above description has been added on page 5.
The results for Figure 1 to 3 and 5 are carried out via simulation. In this simulation, systematic studies were numerically conducted to investigate the optoelectronic characteristics of InGaN-based green µ-LED using the commercial semiconductor device simulation tool Crosslight. The settings of perfectly matched layers (PML) at the lateral and bottom surfaces of the µ-LED were expected to be the most important boundary conditions. The epitaxial structure can be composed of several layers, including a 2.5 μm undoped GaN layer, a 500 nm n-GaN layer with a doping concentration of 3×1023 m-3, five pairs of InGaN/GaN multiple quantum wells (MQWs) as the active region, followed by a 150 nm p-GaN layer with a doping concentration of 2×1024 m-3, and a 100 nm indium tin oxide (ITO) thin film as a current-spreading layer. The composition of the InGaN quantum wells is In0.27Ga0.73N(3.5 nm)/GaN(15 nm), In0.305Ga0.695N(3.5 nm)/GaN(15 nm), and In0.315Ga0.685N(3.5 nm)/GaN(15 nm) for c-plane, semipolar, and m-plane, respectively, resulting in an emission wavelength of 525 nm. The crystal orientation was set to c-plane (0001), semipolar, and m-plane. The above description has been added on page 4.
- The authors point out long-wavelength microLEDs are attractive because of the use in displays. However, it seems like blue microLEDs plus quantum dots for color conversion could be a potential solution to address most issues discussed in the manuscript, such as wavelength shift and wide FWHM. Can the authors discuss this alternative?
Response:
Thank you very much for the comments. Using blue micro-LEDs with quantum dots for color conversion is a potential solution that has been extensively studied. By using blue micro-LEDs and coating them with quantum dots of the desired color, it is possible to achieve a broader range of colors, including red, green, and yellow. This approach is known as a color conversion technique [82, 83]. One of the main advantages of this technique is that it can provide highly efficient and stable color conversion with narrow FWHM, which is essential for display applications [84]. However, there are some challenges to using this approach. One of the main challenges is that quantum dots have limited stability and can degrade over time. Another challenge is that the process of depositing the quantum dots onto the micro-LED surface can be complex and may require careful optimization [83, 85]. The above description has been added on page 20.
- The size effect also introduces sidewall defects that could increase the RC time constant, yet the authors did not mention this at all. Is this not a potential issue for microLED VLC? It seems like the 50 um devices give better LOP than the 30 um devices, this is true even if LOP is normalized by the area. Smaller microLEDs should give better LOP per area than the larger counterpart, why is it not the case here?
Response:
Thank you very much for the comments. For LEDs with sizes smaller than 100 μm x 100 μm, the radiative lifetime becomes the dominant factor in determining the modulation bandwidth, because the geometric capacitance is small enough to prevent the RC time constant from becoming the dominant factor. In addition, the use of atomic layer deposition (ALD) passivation can effectively mitigate the sidewall defects, as demonstrated in our previous research [41,45].
The 30 μm micro-LED array has better light output per area (LOP) than the 50 μm micro-LED array at the same current. At a current density of 5217 A/cm2 and 2500 A/cm2 for the 30 μm micro-LED array and 50 μm micro-LED array, respectively, the corresponding LOP per area is 5.65×10-3 mW/μm2 and 3.39×10-3 mW/μm2 for the 30 μm micro-LED array and 50 μm micro-LED array, respectively, where the current for both device is 0.295 A.
- The authors claimed AlGaInP red microLEDs are not suitable for VLC due to the size effect. What about their optical properties compared with InGaN red LEDs? The authors cited a AlGaInP microLEDs with sidewall treatments, so the mitigation of the size effect in AlGaInP microLEDs is possible. Also, references for ALD passivation and the development of InGaN red LEDs are lacking.
Response:
Thank you very much for the comments. AlGaInP-based red micro-LEDs have demonstrated higher VLC performance; however, they had serious surface recombination problems that cause instability at high temperatures and size-dependent efficiency droop. Due to this reason, InGaN red micro-LEDs are preferred as compared to AlGaInP red micro-LEDs as they can achieve higher modulation bandwidth. Even though the sidewall defects of AlGaInP due to size reduction can be mitigated by using ALD passivation, the existing instability effect and efficiency droop issues limit their potential for high performance VLC applications. The reference for ALD passivation is inserted in the updated manuscript and since it is a review paper, we have not provided the details for the development of InGaN red micro-LEDs, instead we have provided the necessary references.
I would recommend to proofread the manuscript for language use. Some terms and sentences like "... is very relatively rare" and "When the indium content in the InGaN quantum well increases, the QCSE becomes stronger, a phenomenon known as the "green gap" that limits the development of green LEDs" should be corrected to improve readability before resubmission.
Response:
Thank you very much for the comments. We have made necessary changes to the manuscript.

Round 2
Reviewer 2 Report
The authors corrected the article based on most of the comments and observations. It can be accepted.
Reviewer 3 Report
N/A